# First Steps toward Voice User Interfaces for Web-Based Navigation of Geographic Information: A Spanish Terms Study

Teresa Blanco [1,2], Sergio Martín-Segura [3], Juan López de Larrinzar [1], Rubén Béjar [2,3] and Francisco Javier Zarazaga-Soria [2,3,*]

1   GeoSpatium Lab S.L., 50015 Zaragoza, Spain
2   Engineering Research Institute of Aragon (I3A), University of Zaragoza, 50018 Zaragoza, Spain
3   Department of Computer Science and System Engineering, University of Zaragoza, 50018 Zaragoza, Spain
*   Correspondence: javy@unizar.es

**Abstract:** This work presents the first steps toward developing specific technology for voice user interfaces for geographic information systems. Despite having many general elements, such as voice recognition libraries, the current technology still lacks the ability to fully understand and process the semantics that real users apply to command geographic information systems. This paper presents the results of three connected experiments, following a mixed-methods approach. The first experiment focused on identifying the most common words used when working with maps in a web browser. The second experiment developed an understanding of the chain of commands used for map management for a specific objective. Finally, the third experiment involved the development of a prototype to validate this understanding. Using data and fieldwork, we created a minimum corpus of terms in Spanish. In addition, we identified the particularities of use and user profiles to consider in a voice user interface for geographic information systems, involving the user's proprioception concerning the world and technology. These user profiles can be considered in future designs of human–technology interaction products. All the data collected and the source code of the prototype are provided as additional material, free to use and modify.

**Keywords:** voice user interface; geographic information systems; human–computer interaction; user-centered design; web accessibility; methodology; semantics





## 1. Introduction

Voice user interfaces (VUIs) allow voice interactions between devices and people. These interfaces use speech recognition to perceive spoken commands and provide the functionality associated with those commands. Most current VUIs operate under the approach of smart, or intelligent, assistants: systems such as Siri, Cortana, Amazon Alexa, Google Home, and Bixby are novel interfaces that can access various technological devices that surround us, as predicted by Gartner [1]. All of this contributes to the new model of comprehension, utilization, and interaction within the ambient intelligence paradigm, where technology, although almost invisible, is more present than ever [2,3]. An intelligent assistant provides a whole series of technological resources designed to take what we request by voice and convert it into specific tasks. To do this, they employ powerful voice recognition capabilities, then artificial intelligence systems enable them to interpret requests and convert them into commands for our technological devices.

Although these assistants cover many scenarios and much content, related advanced services are lacking in the field of geographic orientation, despite it being an environment of early technology adoption. For instance, methods of user–computer interaction underwent a revolution related to the introduction of graphical user interfaces (GUIs) and their popularization by the Apple Macintosh in the early 1980s. Geographic information systems (GISs) adopted them at an early stage for logical reasons (Figure 1). Now, even

with the help of the GUIs, GISs are inherently complex, and their interactions are often complicated [4]. Therefore, interest has been sustained over recent decades in creating new interaction models, including voice control [5,6]. However, this has not directly translated into practice.

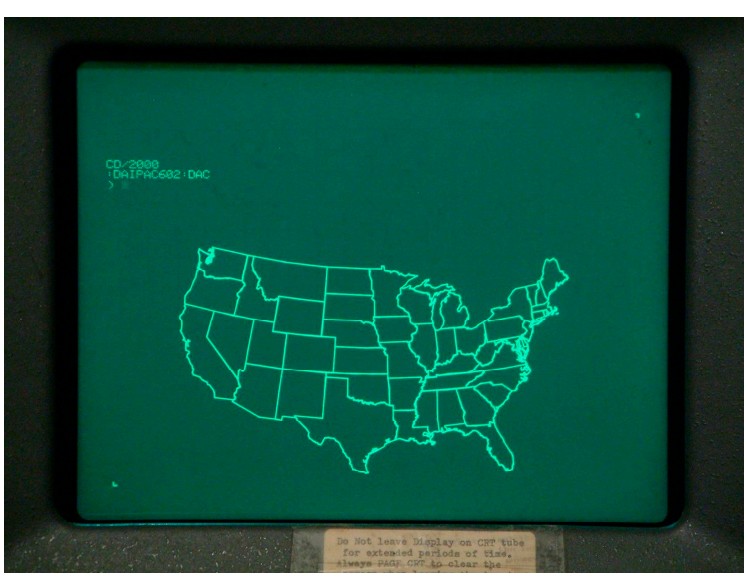

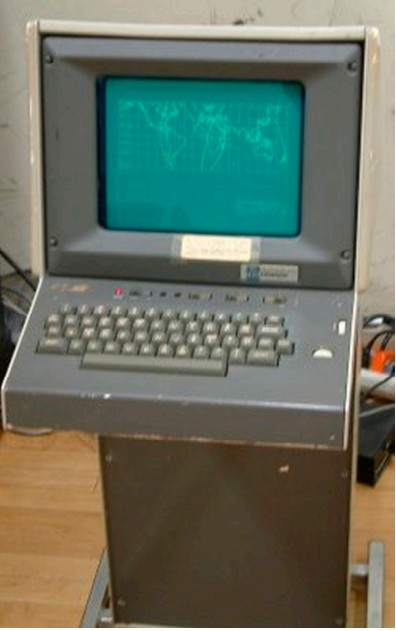

(**a**) Map of the contiguous United States on the Tektronix 4010

(**b**) Tektronix 4010-1 storage-tube-based graphics terminal

**Figure 1.** Example of user interfaces for GIS before GUIs. (**a**) (Reprinted from [7] with permission from David Gesswein/GNUplot); (**b**) (Reprinted from [8] with permission from David Gesswein).

The poor involvement of geographic-based systems in virtual assistance was illustrated in [9]. In 2021, the authors reviewed the state of geospatial applications regarding the use of intelligent virtual assistants. The study emphasized that the use of geographic-based data was built around providing access to gazetteers. Furthermore, [10] presented a dialogue-based mobile virtual assistant that guided tourists to points of interest. It used geographic-based data to present the routes, but the interaction was based on searching for places and providing instructions to find them. These systems are linked to a smartphone-based GIS, and the devices use voice input and output channels in conjunction with precise positioning abilities, mainly through global navigation satellite systems. Nevertheless, these approaches do not provide any kind of interaction with the data presentation (mainly maps).

Google Assistant currently has the highest level of integration with maps and navigation, implementing search options for specific places nearby (e.g., "What street is this?" or "Where are nearby petrol stations?") and managing navigation ("How far am I from . . . ?", "How is the traffic?", or "I want to go to . . . , tell me the next step."). The operating system executes these functions in associated apps such as Google Maps. However, it does not reach levels that emulate a truly natural interaction, unlike interaction models from traditional software, and it has limited functionalities (relying on the quoted questions mentioned above). Meanwhile, the current integration effort in vehicles for hands-free operation of mobile devices is noteworthy [11]. Nevertheless, we found no work in the literature related to using virtual assistants with systems that operate with geographic information beyond browsers.

Another pending issue in the field of GISs is the problem of accessibility regulations and standards, which has been always burdensome in this realm. For example, the public administration's services available to citizens meet very high accessibility standards; how-

ever, maps do not currently meet these standards. In fact, the European Directive on the accessibility of websites and mobile applications offered by the public sector [12] states explicitly, "Article 1.3. This Directive does not apply to the following websites and mobile applications: . . . (d) online maps and mapping services, as long as essential information is provided in an accessible digital manner for maps intended for navigational use." We can also find this translated in other national directives [13].

Nevertheless, the battle is not lost, and we can point to research that evaluated existing limitations and proposed how to overcome them [14–19]. One way to circumvent these access limitations is through other types of interfaces, such as VUIs. Of note, the field of accessibility is not limited to applications for disabilities but covers the entire population spectrum. Integrative orientations, such as Design for All, Universal Design, or Inclusive Design, emphasize that accessibility problems can arise from not only circumstances at birth but also age, illness, trauma, life changes [20–22], or the education level [23]. In line with this, the more recent term "place-ona" [24] (based on the widely known concept of personas [25]) shows how the user's location and activity influence their choice of technology (and, therefore, acceptance). For example, in the car, the place-ona is "eyes busy, hands busy, ears free, voice free"; in the kitchen, it is "hands busy, eyes, ears and voice free" [26].

Furthermore, in the changing technological context illustrated before, designing smart products entails not only the definition of technologies, services, components, and communication processes but also the consideration of new relationships of understanding, using, and interacting between humans and technology [3,22,27]. In this line and according to user-centered design philosophies, understanding user needs and behavior is essential [28,29]. As [22] stated, we should identify the problems users face, view them from their perspective, develop a critical view of the need to define related product functionalities, and humanize the software we create.

Nonetheless, an outstanding issue is establishing effective interaction with the user. For example, the Loopventures test in [30] highlighted this point by assessing the gap between success in understanding (i.e., the extent to which the assistant correctly registers the user's words), which is close to 100%, and success in executing the appropriate response to the user's needs (i.e., whether the response provided is correct). The resulting gap ranged from 52.4% to 85.5%. In other words, a gap exists between the effectiveness of grammar (which works properly) and the establishment of prompts (which fail in many cases). However, even the first level of success has not been achieved for GISs since a corpus of natural language terms is not available that allows the usual functions of a GIS to be commanded.

This paper develops technology elements to gain knowledge of how we can expand VUIs for GISs. This expansion might empower virtual assistants with advanced capabilities so they can deliver more advanced GIS services. A voice-based approach, in particular, may offer tools to improve these systems' accessibility.

The work focuses on managing the interaction with a web-based user interface that can correspond with Google Maps and other geographic information visualization tools offered by spatial data infrastructures [31]. Despite having many general elements, such as voice recognition libraries, the current technology still lacks the ability to fully understand and process the semantics that real users apply to command the geographic information systems. This paper presents a three-part study assessing and improving VUIs for Spanish speaking users, which follows a mixed-methods approach [32]. The first experiment focused on identifying the most common words used when working with maps in a web browser. The second experiment gathered new understanding of the chain of commands used for map management for a specific objective. Finally, the third experiment involved the development of a prototype to validate this understanding. The source code for this prototype is provided as additional material free of charge (File S2). Details about the license for this source code are also provided.

The remainder of the paper is structured as follows. Section 2 presents the materials used in the study, details of the methods followed, and the theoretical background. Section 3 explains the work performed to identify the words usually used to command maps. Then, the process is presented in Section 4, followed by the results obtained when determining the chain of commands used to manage maps. Details relating to the Wizard of Oz experiment and the creation and validation of a prototype are also provided. Section 5 discusses the results, highlighting the particularities of use and user profiles to consider in the VUI for GIS, which involve the user's proprioception regarding technology and the world. Finally, the paper ends with our conclusions.

## 2. Materials and Methods

The experiment was designed and developed using a sequential process in which we connected two assessments with different objectives and strategies, following the Xassess evaluation methodology [32]. We selected Xassess because it is a methodology for user needs and product assessments by multidisciplinary teams (as is the author's team), which offers several advantages: it easily adapts to the context, considers the assessment from the initial stages of the project, merges qualitative and quantitative approaches, and includes the user's vision from the beginning of the project. Xassess may involve three evaluation strategies: complementation (each product dimension is evaluated with one qualitative or quantitative technique), triangulation (each dimension is evaluated with two or more parallel techniques), and combination (each dimension is evaluated with two or more successive techniques). In this study, we followed the combination strategy with a final triangulation, as shown in Table 1.

**Table 1.** Methods, dimensions, and objectives of our methodology.

| Step | Objective | Method | No. of Samples $n$ |
|---|---|---|---|
| 1 | Identify terminological corpus. | Survey. | 137 |
| 2 | Identify the chain of commands. User behavior and acceptance. | Wizard of Oz. Interviews. | 10 |
| 3 | Expert testing before user test. Validate prototype functionality. | Cognitive walkthrough + heuristics. | 3 |
| | Create technology for VUI as a first approach.Validate results obtained in previous steps. | Prototype + user testing. Interviews. | 8 |
| 4 | Emerging issues (transversal). | Triangulation of methods. | |
| | | | 158 (total) |

The first experiment enabled us to identify the main terminology used to manage maps on a web browser. Using the mouse, we are accustomed to zooming in or panning around the map in any direction. However, would we verbally instruct the browser to move the map "to the right" or "to the East"? Initially, we developed a study to identify the common words used to command a map in a browser. To that end, we conducted an online video-based survey with a multimedia questionnaire including mainly open questions, which were answered by 137 Spanish speakers aged 16–75 (Figure 2). The survey was distributed through mailing lists to many users, to gain a diverse sample. For example, although this study was developed in Spain, effort was made to reach other Spanish-speaking countries and sample Latin Americans (we obtained answers from Argentina 1.48%, Colombia 4.44%, Chile 1.48%, Cuba 1.48%, Guatemala 1.48%, México 2.22%, Perú 2.22%, the Dominican Republic 1.48%, and Spain 83.70%). The data are included with this paper in the Supplementary Materials (File S1).

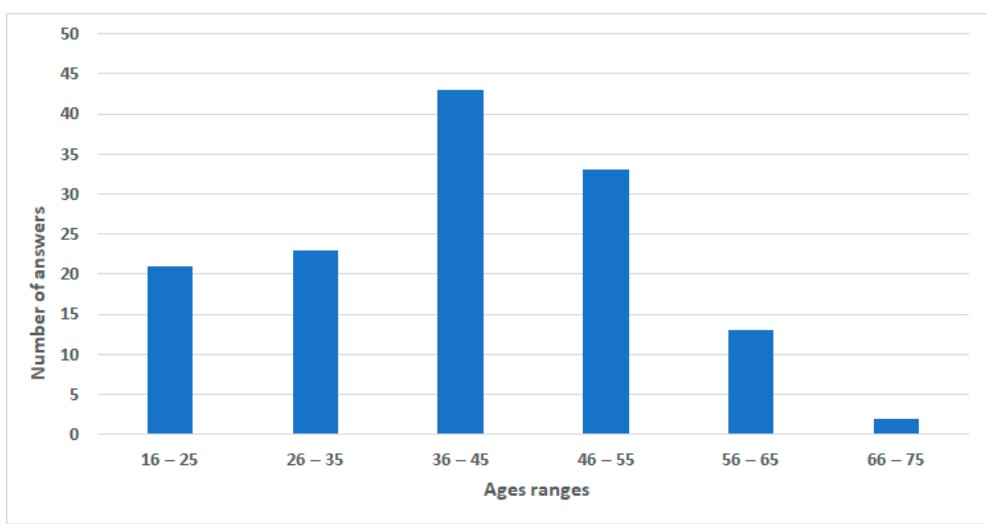

**Figure 2.** Age range of survey respondents.

Based on the survey results and subsequent conclusions, the second experiment aimed to determine how users employ those common terms by creating a chain of commands to yield useful responses from the system. Once the system could understand the main commands, the next challenge was to connect them to provide similar end-user functionality as that obtainable with a mouse and keyboard. For instance, when traveling to Sri Lanka on business, one would probably like to see the tourist sites. How can we develop this particular functionality for the system? Other concerns include how the user will react, how the user will chain commands, and what happens if the system fails to respond properly (the noun Sri Lanka is difficult to pronounce for Spanish people). In the latter part of the second experiment, we carried out a qualitative assessment, using the Wizard of Oz method, with 10 people aged 22–56. The user profile varied in terms of gender, age, and familiarity with technology and GIS systems, to reflect the profiles of the survey population. Due to the in-person nature of the experiment, all of the users were Spanish nationals. Gender parity was achieved and an age distribution similar to that of the survey population (Figure 2): two individuals were 16–25; two were 26–35; three were 36–45; two were 46–55; and one was 56–65. Finally, with the knowledge acquired in Experiments 1 and 2, we developed a working prototype to be evaluated in a final experiment with eight new users aged 20–61. This third experiment reproduced the findings of the second experiment with a real system. Users were recruited according to the same criteria as in Experiment 2: variability in gender, familiarity with technology and GIS, and an age distribution. Before testing a prototype with users, we executed a cognitive walkthrough in combination with a heuristic evaluation [33,34]. In this step, three experts, two senior computer scientists, and one senior geographer performed tasks with the prototype, evaluating it from a specialist perspective and recording each session with screencast software. Each of them possessed over eight years of experience. The objective was to experiment with the prototype's functionality, addressing potential failures and polishing deficiencies before moving on to the final tests with users.

We conducted the assessments in accordance with ethical guidelines, providing a verbal explanation and obtaining written consent from the participants.

## 3. Results

### 3.1. Identifying the Words to Command Maps

For the first experiment, we used a questionnaire to determine the words used in the most common commands to manage a map: zoom in, zoom out, move according to different cardinal points, go to a specific place (a street or a city), rotate the map to the left/right, and change a layer.

The main methodological challenge of this step was not to bias of respondents but instead trying to clearly define what command type of action we needed. For instance, we could not include a question such as, "How do you zoom in?" because the typical response from most people would be to "zoom in". Instead, we selected an approach for dealing with this conditioning by showing videos exemplifying the basic commands. We then asked in open-question form how the survey respondent would request (verbally) that the map to perform these actions (Figure 3). To cluster the answers, we asked about the user's experience with virtual assistants, digital maps' management, and digital competence, in general, in addition to the basic items of user modeling (age, sex, level of education). The level of technological literacy and attitude toward technology are relevant to users' acceptance and interaction with new technological products [35–37].

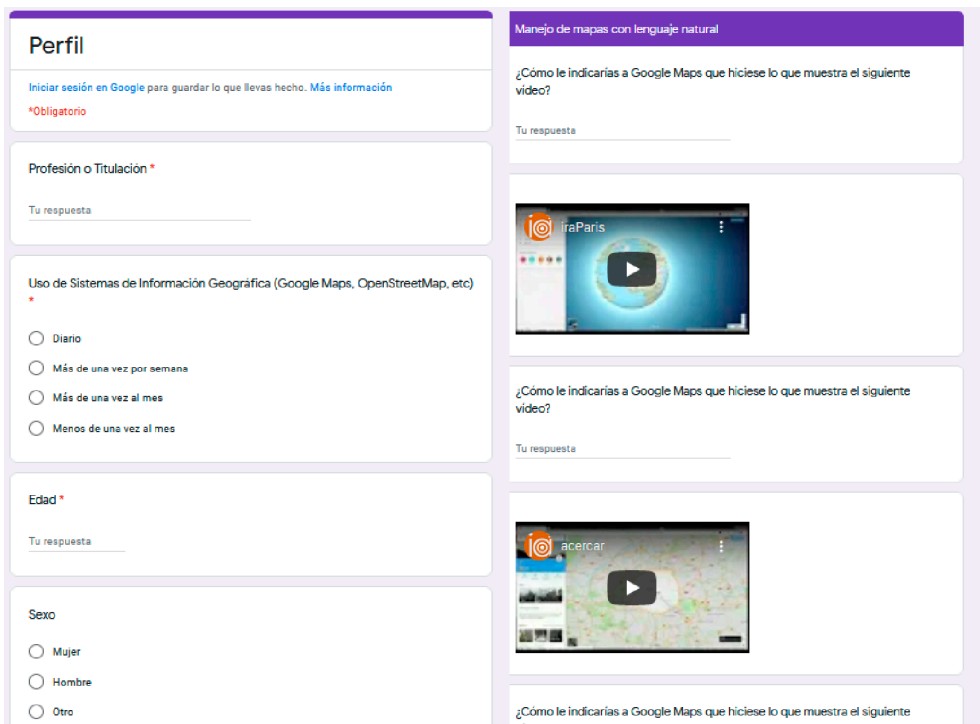

**Figure 3.** Questionnaire using videos.

The collected results were sorted by the verbs, conjugation modes, and nouns used. Given the variety of responses in terms of spelling, abbreviations, and conjugations, several iterations of sorting were developed. We employed tools including TableauPublic (free SaaS version), Python's Pandas Database library, and the Notepad++ word processor to process the first automatic classifications. Then, we conducted two card-sorting [38] sessions with three experts from different disciplines (information, geography, and design). First, open card sorting was developed individually and blindly; second, hybrid card sorting was developed through a group dynamics session aimed at pooling and obtaining agreement between the categories determined by each expert and the machine. Once fully classified, the data were tallied and processed to identify profiles and emerging patterns between the vocabulary and user characteristics.

The asymmetrical distribution that we obtained provided very informative results: despite the large variety of vocabulary, words, and synonyms mentioned before, the most frequent answers (in some cases 95%) used one of only three specific words. As shown in Figure 4, the most frequent verbs used to move the screen by voice were *desplazar* (to move), *mover* (to move), *ir* (to go), and the directional verbs *subir* (to move up) and *bajar* (to move down). In the analysis, both directional verbs were combined because the use of either was directly related to the action presented in the video. Our grouping included

their infinitive and imperative modes. Often, when conjugated in the imperative, the verb is accompanied by a pronominal particle (e.g., "move yourself" or "move me").

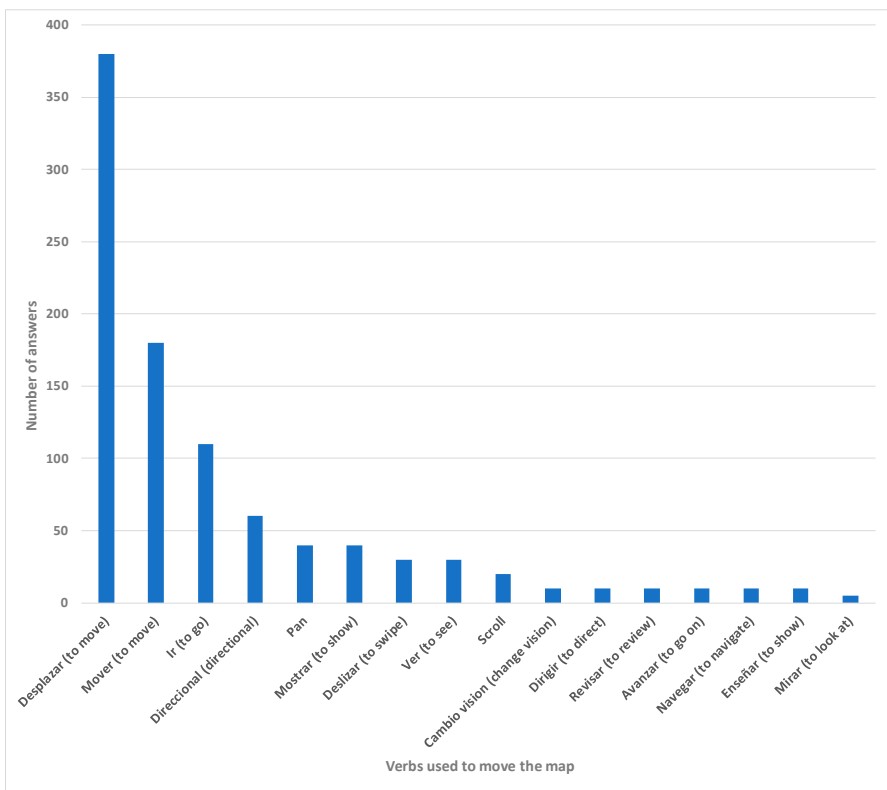

**Figure 4.** Verbs used to move the map.

As anticipated, words for the zoom-in action had little variability, with only seven different verbs, three of which were used by 95% of users (Figure 5). The most frequently used verbs related to this action were *ampliar* (to enlarge), *acercar* (to zoom in), and "zoom" (Spanish Anglicism for zoom in, frequently used with the verb *hacer*). *Ampliar* and *acercar* can be conjugated in the second person imperative and accompanied by a direct complement "map" or "image", but the infinitive is more common.

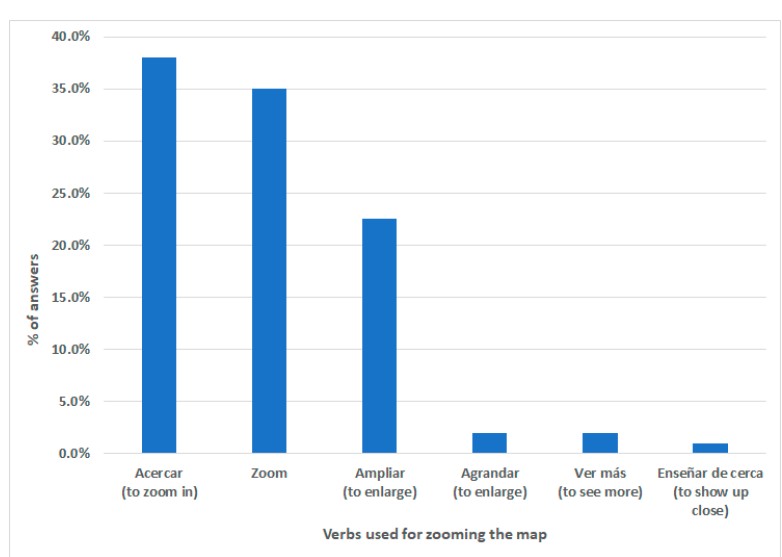

**Figure 5.** Verbs used for zooming in the map.

Figure 6 shows the most commonly used verbs to display an orthophoto as an example of loading a new layer. They were *cambiar* (to change), *mostrar* (to display), *ver* (to view), and *vista* (view as a noun). Of note, the use of the term "satellite" occurred both as a noun with other instructions and as a complete instruction (without using a verb, "*No verbo*" in Figure 6).

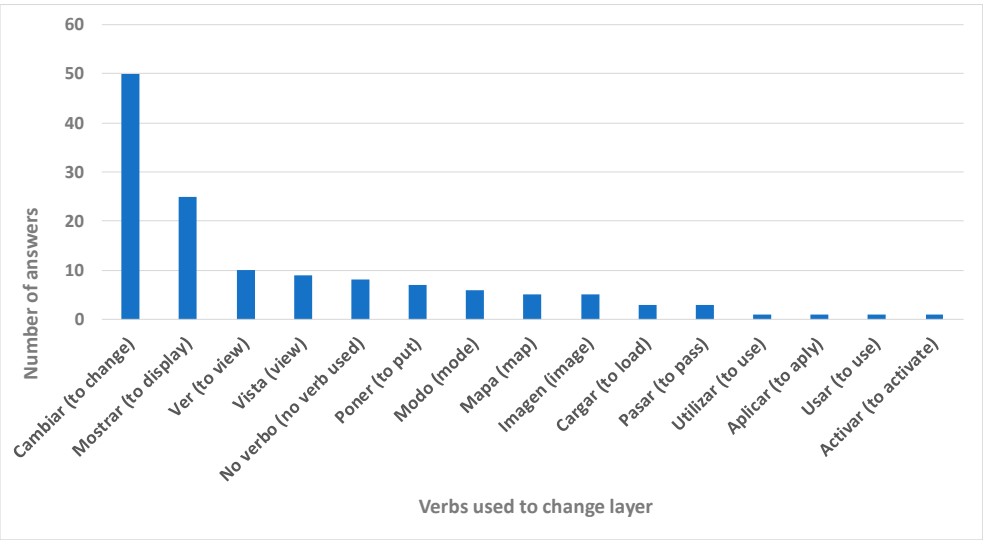

**Figure 6.** Requests to change layer functionality.

All correlations between age, gender, level of education, and level of familiarity with virtual assistants or GISs and differences in vocabularies were analyzed. The results were largely consistent within each category. However, the untrained, elderly, and children were too poorly represented to determine correlations for these user profiles with any certainty. Additional dedicated studies with more users in these specific groups are needed to observe their specific characteristics.

The most important result of the survey was the identification of a corpus of vocabulary used to interact by voice with a map.

### 3.2. Identifying the Chain of Commands Used

The survey provided information about interaction phrasing but not the chain of commands used to complete specific tasks. To observe the approach and strategy, as well as the response to errors, that users would adopt in a real case, a Wizard of Oz experiment was employed. Using the results of this experiment, we built an initial prototype as a first approach to creating technology for a VUI, as well as to validate the results of the Wizard of Oz experiment.

#### 3.2.1. Wizard of Oz Experiment

In a Wizard of Oz [39] experiment, users interact with a simulated system, the name of which comes from the novel in which a human hidden behind a screen creates the effect of something that does not exist. In this case, a system was simulated to perfectly interpret the users' natural language and perform the required actions on the GIS, as the user's screen was connected to the wizard's computer. The "wizard", who was in the same room, hidden behind a wall, could hear what the user was saying (Figure 7). The experiment was carried out by three researchers, who rotated among two predetermined roles: the interviewer and the wizard. The former greeted users at the door, explained the general objective of the test, and led the user to the test station adjacent to the wizard's hideout. Then, they observed the test performance and took field notes, without interfering with the user–system interaction.

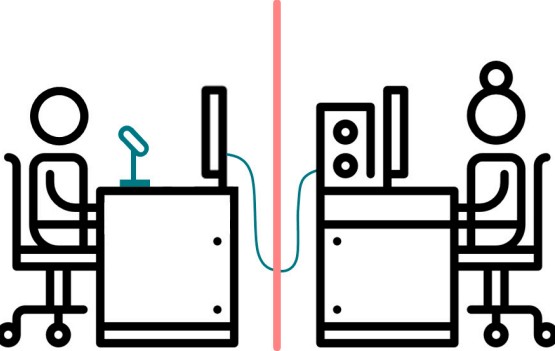

**Figure 7.** Configuration of the scenario for the Wizard of Oz experiment (figure from the authors, designed using resources from [40]).

As shown in Table 2, four scenarios were designed for the users to run through all the typical map interaction functionalities. These scenarios naturally integrated various necessities of functionality but also induced errors to observe the full range of user behavior in different situations. For each scenario, a list of predicted errors and a set of search terms were developed to allow the wizard to react to users with enough speed to be convincing. For this purpose, strategies were specified to mask the use of peripherals and sound, all to maintain the illusion and thus elicit natural reactions from the users. A brief satisfaction interview followed (asking about difficulties observed during the test, their perceptions, and other emerging issues) and an anonymized user profile survey. Then, the deception was revealed, and we offered users the option for us to remove the recording of the experiment from our study if they wanted. Finally, we spoke with each user to ascertain their reactions to understanding the whole experiment.

**Table 2.** Wizard of Oz Scenarios, Predicted Errors, and Error Mitigation.

| Scenario Description | Planned Errors | Planned Searches for the Wizard |
|---|---|---|
| Your Aunt Lisa is on a business trip to Sri Lanka and you would like to know about the island. Locate the island and research the capital city (Colombo) and national parks. | The difficulty of pronouncing the name of the island. | Colombo, Kandy, Jaffna, Batticaloa, Galle, Srilanka, Sri Lanka, India, Asia |
| You are looking for a park in Paris to have a picnic. It is summer, and you want a place with plenty of trees and shade. Compare the leafiness of the parks using satellite images. | If you look for *parque París* (Paris park) in Google Maps from an IP in Spain, the first answer is Paris Park in Madrid. | París, París Francia, Parque París, París parque, Parque, Bois de Bolougne, Bois de Vincennes |
| You have an appointment with a friend on the first roundabout of the city of Vitoria (Spain), arriving via highway A-132. Look for this point on the map to learn how to get there. | The command *rotonda en Vitoria* (roundabout in Vitoria) sends users to a street with Vitoria in its name (depending on the geographic context of the user).<br>The command *rotonda A-132* shows only the road. | Vitoria, A132, A132 Vitoria, Vitoria A132 |
| You have to carry a package from the Grancasa shopping center in Zaragoza to Pilar Square. Find the pedestrian route without using the routing functionality provided by Google Maps. | You need to zoom in to see the names of the streets. | Grancasa, Centro Comercial Grancasa, CC Grancasa, Calle Gertrudis Gómez de Avellaneda, Gertrudis Gómez de Avellaneda, Calle Gómez de Avellaneda, Gómez de Avellaneda, Calle María Zambrano, María Zambrano |

In selecting users, we sought to achieve diversity in the factors relevant to interacting with systems by voice (e.g., age, experience with computer systems, dialect variety, and level of education). Because of the qualitative character of the methodology, along with the relatively small sample, each recording was studied in detail. Each functionality

that the user tried to engage, and whether the wizard could respond to the request or not, was recorded to determine the optimal form of natural interaction on which to base the prototype.

The main goal of this experiment was to observe the strategy the users employed to solve the problems presented. The results revealed two completely different and opposing profiles. The primary strategy was using the search functionality to solve the problem in a single step. These users stuck with this strategy even after several consecutive searches had failed to resolve the scenario. On the other hand, a minority of users, less versed in interacting with computer systems, including GISs and virtual assistants, tried to solve problems by moving and zooming iteratively.

The observed interactions largely corroborated the vocabulary identified in the survey. Certain critical factors were also detected with importance for the development of the prototype, such as explicit feedback and context persistence. Furthermore, although rare, attempts were made to use elements external to the GIS by voice ("see photos" and "print").

### 3.2.2. Prototype Creation and Validation

Using the results of the Wizard of Oz experiment and the survey, a system prototype was developed to help design technology for a VUI and to collect results in a near-to-reality scenario. This prototype consisted of three parts. First, a server was needed to receive and respond to requests for processing instructions and store the interactions in a database. This was developed using Spring Boot and used an h2 database. Second, a package was devised to process the instructions in natural language and create the appropriate answer in each case. In this prototype, the package was built with JavaCC, a parser generator, using the specifications obtained from the first and second experiments. In the third and final part, the client would oversee the map display provided by WebGL Earth (Figure 8), an open-source virtual planet application, and send requests to the server with the text recognized by the Annyang library, which exposes the functionalities of the WebSpeech API interface. Web Speech API is a JavaScript interface supported by most browsers for speech recognition and synthesis.

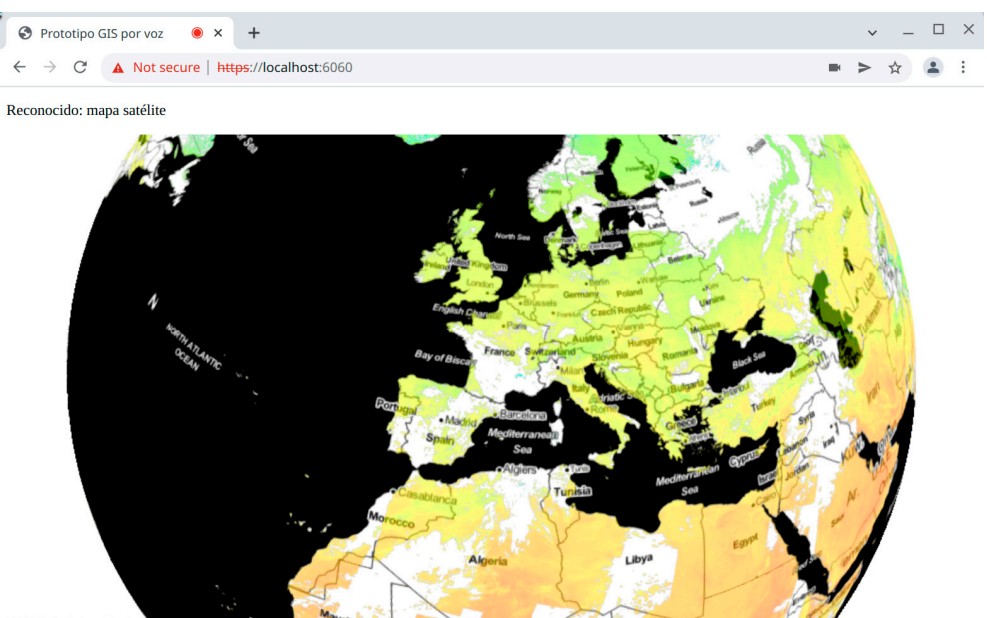

**Figure 8.** Prototype interface.

The prototype could respond to instructions and operate like any map browser using voice alone. It could provide feedback in the form of the recognized text, which was essential in case the system confused instructions for similar words (e.g., nine and move). It could also search for toponyms through the MapQuest API, but correct toponym recognition

could be challenging in other languages or with difficult or variable pronunciation. A video demo showing the prototype in use is provided as additional material (File S3).

The prototype was validated to determine if it could fulfil the requirements of the previous experiments (survey, and Wizard of Oz + interviews) through another scenario experiment based on the Wizard of Oz tasks. As those tasks covered the functionalities comprehensively, the prototype test results could be compared with the ideal system simulated by a human successfully to validate the prototype according to the original specifications. As in the previous experiment, observer intervention was minimized to replicate a natural interaction with the system. The real prototype in operation allowed for interactions to be automatically recorded, facilitating various measures we carried out, such as calculating the success rate of the instructions. The COVID-19 pandemic constrained the experiment in part, forcing testing to be performed online, with a sample of eight. This approach limited our ability to observe during the experiment, but we countered this by using two communication tools simultaneously, one for the user's webcam and another for the user's screen.

The validation was successful, confirming the fixed requirements of the previous experiments were met. When implementing only the most popular verbs for actions, we observed that, in the case of a failure due to an unrecognized term, users quickly switched to a more frequently used term. Only one user of those interviewed expressed that he would not use it in his daily life, and the average rating was 3.8 out of 5.

The source code for the prototype is provided as additional material, free to use and modify (File S2).

## 4. Discussion

The Hype Cycle proposed by Gartner in 2018 [1] predicted that the use of virtual assistants as a technology would plateau in productivity in the next 2–5 years. Five years later, smart assistants are part of our day-to-day. Nevertheless, they have several limitations for managing certain domains. While the devices are capable (designed to understand the voice and transform it into character strings accurately), the information systems connected to them need to evolve to give users access to their functionality via this "new door". GISs, in general, and web map services, in particular, represent a domain that requires growth. This domain is included in several exceptions of national and international directives regarding human–computer interaction, as mentioned in the introduction, and its development seems a challenging task.

In that context, research work was needed to create and share a corpus that will help the industry develop these capabilities for the GISs. Furthermore, we needed to devise specifications for the human–computer interaction process that align with users' needs for their smart assistants. This paper presents several connected experiments as the first step to creating a VUI to manage maps in a GIS. The initial work focuses on understanding how people would like to manage maps in a web browser in Spanish. The specification of the language was demonstrated to be crucial in this domain (and likely in others) because Spanish is a language rich in synonyms. Moreover, Spanish speakers tend to mix Spanish with English terms (e.g., zoom, pan).

An encouraging result of this work is that the vocabulary used in the interaction is easy to narrow down. All functionalities could be used with less than five possible verb options, albeit in different conjugations, to cover more than 95% of the uses. Even in the remaining cases, users quickly switched to more common options after the first failure. However, this does not free us from the need for a machine learning system in the final version; the interpretation of ambiguous instructions and some aspects of context expected by users would be difficult to implement otherwise.

Feedback was also identified as a key element in this type of interaction. The absence of sensory feedback on the user's action (e.g., the click of the mouse or the noise of the keyboard) increased the user's anxiety about whether their action was registered. Appropriate feedback would not only alert the user to repeat their action because their

instruction was misunderstood but also tell the user how it was misunderstood so that they could change their enunciation to avoid successive errors.

One noteworthy finding was the significant influence that Google Maps already had on the way users interacted with this type of product. Their vocabulary was conditioned by which functions are available in Google Maps and how they are named; likewise, we noted that Google Maps has conditioned users' strategies for solving tasks. The strategy followed by most users was the only one currently offered by Google Maps via voice, i.e., the one-step search. In the experiment, we considered this and engineered several scenarios so they could not be resolved with a one-step search. In this situation, some users tried alternative strategies, while others refined the initial order of their strategy iteratively as if they were using a keyboard and mouse.

Of note, some exceptions were evident. Some users did not use the search function and operated as if they had physical maps. This resulted in two user groups (one with the majority of users) that were very disjoint from each other, using significantly different commands. The two groups were not defined by gender or age but by their daily experience of using computer systems (including GISs). This was another case where current technology could strongly influence the interaction with future systems: users had already internalized the capabilities and limitations of existing GISs.

In this spirit, after triangulating the results of the experimentation, we could observe an interesting phenomenon: the different ways in which a user interpreted the system involved the proprioception the user has of the self with respect to technology or the world. This corroborated and complemented the initial segmentation related to technological literacy and previous motivation for the use of technology and allowed us to re-segment the user in the following categories (Figure 9):

- Me in the world. The user locates themself in the world represented by the system, maybe flying. They use expressions such as "go to the north, to the west", "get me closer", and "move me".
- Me on the street. The user imagines themself walking on the ground. They use expressions like "turn right", "move away", "go straight ahead", and "how to go to".
- Me in front of a screen. The user sees the system as a system and does not try to interact naturally but tries to find out how to provoke the desired response. The user interprets the system as an interactive map ("move it"), and the most differentiating factor is that they use languages learned from other systems. This may happen for the following reasons. The user is very connected to the mobile web, app, etc. They use expressions such as "back" (referring to a previous screen or back button), "reverse zoom", or, in the case of a user who directed the cursor orthogonally, "up-up-right-right-up-up" to traverse a diagonal. Another possible reason derives from a subgroup of users with a technical profile, e.g., geographers, computer scientists, and engineers, that prompts them to speak to the system in a programming language or with technical words (e.g., "scroll" or "pan"). In these cases, the user adopts non-conjugated and non-personalized expressions.

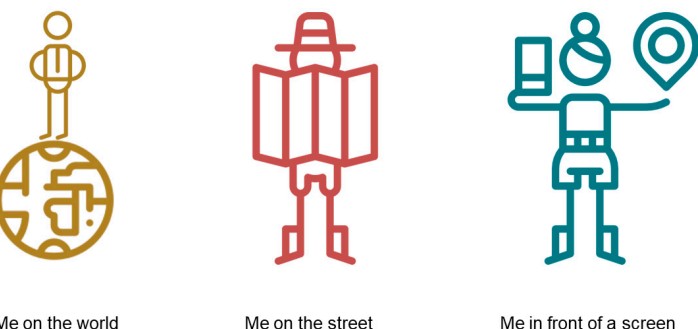

Me on the world          Me on the street          Me in front of a screen

**Figure 9.** User profiles based on their proprioceptions and interactions with the system (figure is from the authors, designed using resources from [41]).

Finally, from the prototype evaluation, we concluded that such a system is a real need. When reviewing the prototype's quality, most users expressed that they would use the tool daily.

However, much work remains in this area. This experimentation methodology allowed us to detect and reflect on issues that require further research and development:

- Voice recognition improvement. The recognition of the moment the user initiates the instruction is crucial. Currently this is solved with the keywords the user must say before verbalizing the request, e.g., the assistant's name. In addition, the transitions between instructions require enhancement. Another issue is the understanding of phonetically similar words. For example, the user says *mueve* ("move"), and the system might interpret it as *nueve* ("nine"). This is understandable and can perhaps be solved by limiting the vocabulary the system recognizes. Most speech recognizers use all the vocabulary of the language in which they are executed, and many misunderstandings occur with words that are not used in the GIS context. When similar words are necessary for a certain context, as in a GIS (e.g., *mueve* and *nueve*), a feasible solution is to improve the voice recognition software. Finally, the recognition software can be enhanced to achieve comfortable use without requiring loud voice commands (e.g., when outdoors). In addition, we recommend using high-quality microphones, such as the ones used in our experiments.

- Toponym problem. Most words and speech recognizers use the vocabulary of the language being spoken. The problem is that toponyms do not always belong to that language. Some have several accepted pronunciations, and users pronounce them differently, whether accepted or not. Therefore, a speech recognizer capable of detecting the correct toponym being pronounced and selecting the appropriate one from the endless list of possible toponyms is a challenge to design.

- Persistence of the context. In the developed prototype, if a user asks the map to move to the right, the system moves it to the right a certain distance, and if the user makes the same request again, the map moves that same distance. To improve the user experience significantly, the context can be addressed such that if the user asks to move to the right and then says "further" or "further to the right," the map moves a distance larger than the first. In other words, the system can respond according to prior commands. Context persistence and machine learning can also teach the system iteratively based on the user's pronunciation and vocabulary. As a result, the product can better adapt to the range of possible users (considering pronunciation and the different user profiles previously mentioned), improving the user experience.

- More user studies. Additional user studies should be conducted to observe the behavior and attitudes of the different profiles in different place-onas. The groups at risk of isolation in the spoken interaction paradigm should also be considered. Specifically, we need to address the specific difficulties faced by people with social communication disorders, such as dyslalia, dysphemia, dysarthria, and dysglossia, and to what extent they can make use of speech recognition standards. Similarly, for people with aphasia or cognitive disabilities, specific solutions can allow them to access these systems. Finally, the vocabulary and interaction of children and the elderly with GISs should be studied.

## 5. Conclusions

Gartner's 2018 Hype Cycle predicted that virtual assistants would reach a plateau of productivity; five years later, they have become a common feature in daily life as a "new door" to technology but still have limitations on their effectiveness in certain domains. One of these domains is geographic information systems (GISs), specifically web map services, which face difficulties in understanding and handling the semantics or meaning and context behind the commands inputted by users, preventing the technology from effectively processing users' requests and providing accurate results. In that context, it was necessary to conduct research to collect data that can aid in the enhancement of human–

computer interaction for GISs and to establish guidelines for human–computer interaction that take into account user needs for smart assistants in this field.

This paper presents our initial efforts to create specialized technology for voice user interfaces in geographic information systems, using Spanish as the primary language. Three connected experiments were conducted using a mixed-methods approach to examine the common words used in web-based map navigation, to analyze the sequence of commands used for specific map management tasks, and to understand the particularities of user interaction in this scenario. Based on the previous experiments, we developed a minimum corpus of Spanish terms and constructed a prototype to validate the findings.

Due to the complexities of the Spanish language, we encountered a large number of possible interaction terms for certain commands. However, it was found that the vocabulary could be easily narrowed down as all functionalities required less than five possible verb options to cover more than 95% of uses. Some key issues were identified with this type of interaction: feedback is a key element as its absence increases user anxiety; the daily experience of the user with technological products, and particularly with GIS, influences the use of terms and user behavior, as users have internalized the capabilities and limitations of existing technologies. This influence is greater than age or gender; therefore, it is necessary to take this into account in the design of technological products' interaction capabilities.

Additionally, and related to the above, an interesting conceptual finding was that the ways in which a user interprets the system involve the proprioception the user has of themselves in relation to the technology or the world. Based on this, three typical user profiles were defined that can be considered in future designs of human–technology interaction products. The article also presented use cases that can be considered for voice user interfaces (VUIs) in geographic information systems (GIS).

As we have found out, such a system represents a response to a real need. It is both wanted and accepted by users. However, much work remains to be done in this field, especially in terms of voice recognition, toponym management, consideration of the context, and user studies including groups at risk of isolation. Furthermore, this work has been carried out using a web-based interface; it should be developed by analyzing the interaction with a smartphone-based interface to determine if the user behavior is the same or if the terms used are different.

**Supplementary Materials:** The following supporting information can be downloaded at https://github.com/IAAA-Lab/VUI_Prototype (Retrieved 30 December 2022): File S1: Data collected during the survey, https://github.com/IAAA-Lab/VUI_Prototype/releases/download/v0.1.0/dataSurvey.rar. File S2: Source code for the prototype, https://github.com/IAAA-Lab/VUI_Prototype/archive/refs/tags/v0.1.0.zip. File S3. Demo video of the use of the prototype, https://github.com/IAAA-Lab/VUI_Prototype/releases/download/v0.1.0/video-demo.mp4.

**Author Contributions:** Conceptualization, T.B. and F.J.Z.-S.; methodology, T.B. and F.J.Z.-S.; software, S.M.-S. and J.L.d.L.; validation, T.B. and R.B.; formal analysis, T.B. and F.J.Z.-S.; investigation, T.B.; resources, F.J.Z.-S.; data curation, T.B., J.L.d.L., and R.B.; writing—original draft preparation, T.B. and F.J.Z.-S.; writing—review and editing, S.M.-S. and R.B.; visualization, S.M.-S. and R.B.; supervision, F.J.Z.-S.; project administration, J.L.d.L. and R.B; funding acquisition, T.B. and F.J.Z.-S. All authors have read and agreed to the published version of the manuscript.

**Funding:** This paper is part of the R&D project PID2020-113353RB-I00, supported by the Spanish MCIN/AEI/10.13039/501100011033, and the project T59_20R, supported by the Aragon regional government. The work of Teresa Blanco is part of the project PTQ2018-010045, supported by the Spanish government program Torres Quevedo.

**Institutional Review Board Statement:** The paper was reviewed by the Research Ethics Committee of the Community of Aragón (CEICA) in Spain. This Committee considered that realization of the article had not violated any ethical or legal principle, had obtained the corresponding informed consent, and complied with current legislation on the protection of personal data.

**Informed Consent Statement:** Informed consent was obtained from all subjects involved in the study.

**Data Availability Statement:** The shareable research data can be found on the links in the Supplementary Materials section.

**Acknowledgments:** We thank Guillermo Reloba for his collaboration with parts of this work. We also thank David Gesswein for the authorization for using the pictures presented in Figure 1.

**Conflicts of Interest:** The authors declare no conflict of interest. The funders had no role in the design of the study; in the collection, analyses, or interpretation of data; in the writing of the manuscript; or in the decision to publish the results.

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
