# Peer review of "First Steps toward Voice User Interfaces for Web-Based Navigation of Geographic Information: A Spanish Terms Study"

_applsci, doi:10.3390/app13042083_

Round 1

Reviewer 1 Report

The subject of this paper is very interesting and well written. Therefore, it is suitable for 'Applied Sciences'.

Author Response

Thanks for your comments. We are pleased that you found our work interesting.

Reviewer 2 Report

-Figure 3,4,5,6,8 => Overall, the text is not visible in these figures cture because it is too blurry. Picture quality needs to be improved.

-Figure 4,5,6 => In the spanishi words at the bottom, English should be written together so that readers can understand.

- Google Maps,used as an example in this study, is a kind of web mapping or navigation for general users. It can be seen as distinct from traditional GIS such as ArcGIS or QGIS used for spatial analysis such as urban planning. Therefore, VUI for Web Mapping or Navigation seems more appropriate than VUI for GIS in the title.

- In the case of Google Maps, it is used more as an app on a smartphone than a PC.  In the case of a smartphone, a basic VUI is provided. So, it seems more appropriate to conduct a case study in the environment of a smartphone rather than a PC. It is necessary to describe what was performed in the PC environment rather than mobile as a limitation of this study or a future improvement point.

- In the case of step 1, the sufficient number of samples and descriptions of the samples are appropriately described, but in the case of step 2 and step 3, there is insufficient explanation of what criteria each sample was selected for. It is necessary to explain what kind of people the samples in step 2 and 3 were and how they were selected.

Author Response

We appreciate your comments. Following, we are providing response to them.

-Figure 3,4,5,6,8 => Overall, the text is not visible in these figures cture because it is too blurry. Picture quality needs to be improved.

We have improved the images but there is a limitation regarding with the space in the template. Nevertheless, you could click in the online version and see a bigger image.

 -Figure 4,5,6 => In the spanishi words at the bottom, English should be written together so that readers can understand.

Done

 - Google Maps,used as an example in this study, is a kind of web mapping or navigation for general users. It can be seen as distinct from traditional GIS such as ArcGIS or QGIS used for spatial analysis such as urban planning. Therefore, VUI for Web Mapping or Navigation seems more appropriate than VUI for GIS in the title.

We have changed the title according with your comment and other provided by other reviewer. Our new proposal is: First steps toward voice user interfaces for web-based navigation of geographic information. A Spanish terms study

- In the case of Google Maps, it is used more as an app on a smartphone than a PC.  In the case of a smartphone, a basic VUI is provided. So, it seems more appropriate to conduct a case study in the environment of a smartphone rather than a PC. It is necessary to describe what was performed in the PC environment rather than mobile as a limitation of this study or a future improvement point.

We have pointed this comment in the conclusions (third paragraph)

- In the case of step 1, the sufficient number of samples and descriptions of the samples are appropriately described, but in the case of step 2 and step 3, there is insufficient explanation of what criteria each sample was selected for. It is necessary to explain what kind of people the samples in step 2 and 3 were and how they were selected.

We have added more detailed information in lines 181-188, 191-197

Reviewer 3 Report

The manuscript presented to me for review presents very interesting issues. Its main value is not the novelty of the ideas and methods or the computational technology, but the built procedure of thematic specification of the voice search tool. With the subject matter taken into account being Geographic Information Systems (GIS).  I think the article is well constructed. The chapters are arranged appropriately and the procedure is described clearly and with great care.

A few comments that I would like to point out in this review may improve the quality of the manuscript and further facilitate its reception by the reader.

1. Please consider whether the GIS included in the title is not too broad a term at this stage of the research. The experiments and prototype are essentially about working with an interactive map. GIS is a much broader term that includes both thematic cartography, analysis tools, etc.

2. In line 222 the authors write about the study of correlation and its high results. Why don't you give the obtained values?

3. In line 223 the authors write about "the distribution in every case was very skewed". Does it refer to the graphs of fig. 4-6. You can not talk about skew distribution when there are descriptive values on the horizontal axis that do not reflect the severity of the phenomenon. In such a case, the arrangement of values results only from the setting of descriptive values.

I make a similar point with regard to the exponential distribution described in line 256.

4.Conclusions should include a reference to the realized objective and basic results. A large part of them in their present form should be in the discussion. The discussion allows you to refer to existing solutions and problems.

5. Figures 3 through 6 should include an English translation as an alternative. This will make it easier to understand the use and citation of the research by international readers.

Author Response

We appreciate your comments. Following, we are providing response to them.

1. Please consider whether the GIS included in the title is not too broad a term at this stage of the research. The experiments and prototype are essentially about working with an interactive map. GIS is a much broader term that includes both thematic cartography, analysis tools, etc.

We have changed the title according with your comment and other provided by other reviewer. Our new proposal is: First steps toward voice user interfaces for web-based navigation of geographic information. A Spanish terms study

2. In line 222 the authors write about the study of correlation and its high results. Why don't you give the obtained values?

Thank you for bringing this issue to our attention:  we have now realized that there was an error in the references in the text that we had overlooked. There are 38 references, but in the previous text, there were 40. Derived from this, he sentence regarding the chi-square test is also an error: as we explained on line 230, the correlation between the machine and the experts panel was made through a group dynamics (second card sorting). We apologize for the oversight and have corrected the references: [35-36] instead of [35-37] (line 218); [37] instead of [38] (line 226); deleted reference [39]; and [38] instead of [40] (line 274). Additionally, we have slightly corrected the text to improve clarity.

3. In line 223 the authors write about "the distribution in every case was very skewed". Does it refer to the graphs of fig. 4-6. You can not talk about skew distribution when there are descriptive values on the horizontal axis that do not reflect the severity of the phenomenon. In such a case, the arrangement of values results only from the setting of descriptive values.
I make a similar point with regard to the exponential distribution described in line 256.

We believe that this may have been a problem with the translation. When we referred to a "skew distribution," we intended to convey that the distribution was not symmetric. We have revised the terminology used to improve clarity (line 233). In line 265, we aimed to express the same concept, but as it was already stated in line 233, we have removed the sentence for clarity and consistency.

4.Conclusions should include a reference to the realized objective and basic results. A large part of them in their present form should be in the discussion. The discussion allows you to refer to existing solutions and problems.

We have implemented all of your suggestions in the Conclusions and Discussion sections. We hope that you find them accurate.

5. Figures 3 through 6 should include an English translation as an alternative. This will make it easier to understand the use and citation of the research by international readers.

Done in figures 4 through 6. The problem with figure 3 is that it is the Google form already used (and it was in Spanish). Previous paragraph gives a general view of the information the form was collecting. If you consider that this is not enough, we could include an additional paragraph for providing additional details of the information collected.